# Cross-spectral fusion of thermal and RGB imaging for objective pain estimation

**Oussama El Othmani[1,2]☯, Sami Naouali[3]☯***

**1** Computer Science Department, Military Academy of Fondouk Jedid, Nabeul, Tunisia, **2** Military Research Center, Aouina, Tunisia, **3** Information Systems Department, College of Computer Science and Information Technology, King Faisal University, Al Ahsa, Saudi Arabia

☯ These authors contributed equally to this work.
* salnawali@kfu.edu.sa

## Abstract

Pain assessment remains challenging for patients unable to verbally communicate, including neonates, cognitively impaired individuals, sedated patients, and those who suppress expressions due to cultural norms or stoicism. We demonstrate that integrating thermal imaging with RGB facial expression analysis provides more accurate and robust pain intensity estimation than either modality alone. Our dual-camera system records synchronized thermal and RGB video, processed through a cross-spectral attention fusion (CSAF) model with a temporal transformer for continuous 0–10 scale pain prediction. In a controlled laboratory pain induction study, 50 healthy adults (ages 21–68, 87.3 h video) underwent Cold Pressor Test and pressure algometry protocols; our system achieves MAE = 0.79, representing a 33.1% improvement over RGB-only (MAE = 1.18, $p < 0.001$). In a real-world clinical postoperative monitoring study, 30 surgical patients (ages 31–74, 17.7 h video) recovering from abdominal surgery were monitored; our system achieves MAE = 1.08, representing a 28.5% improvement over RGB-only (MAE = 1.51, $p < 0.001$). Across the combined cohort ($n = 80$), MAE = 0.87 (29.3% overall improvement over RGB-only). Benefits increase at higher pain intensities (38.5% at severe pain) and for challenging populations where expressions are suppressed (37.6% for low expressers). Thermal responses precede visible expressions by 1.2 ± 0.3 seconds, enabling earlier detection. This work was validated on adults only; pediatric applications require dedicated validation. Translation to clinical practice requires multi-site prospective trials, regulatory approval, and careful implementation planning.

## Author summary

Pain assessment in patients who cannot speak—newborns, cognitively impaired individuals, or those under sedation—remains one of medicine's most difficult challenges. Clinicians currently rely on observational tools that are time-consuming, subjective, and only applied at discrete time points, potentially missing pain episodes between assessments.

provided the original author and source are credited.

**Data availability statement:** Code and Model Weights: All code and trained model weights underlying the findings of this study is publicly available without restriction at https://github.com/oussama123-ai/Cross-Spectral-Fusion-of-Thermal Anonymized Processed Features, Metadata, and Evaluation Protocol: Deposited as Supporting Information files (Table B and Table C in S1 Appendix) freely downloadable with the published article. These include per-fold cross-validation results, aggregated demographic statistics, pain intensity distributions, and exact train/validation/test split definitions. Raw Data Sample: A representative sample of anonymized raw data underlying this study is publicly deposited at https://zenodo.org/records/18991937 (DOI: 10.5281/zenodo.18991937). This deposit includes example feature arrays to substantiate the processed features described in Table B and Table C in S1 Appendix, in compliance with PLOS open data requirements. Restricted Data: Full raw video recordings cannot be publicly released due to patient privacy protections and institutional review board restrictions (IRB-MRC-MHT-2022-001); this restriction is consistent with the PLOS Digital Health exception policy for ethically restricted data [40]. Requests for access to additional processed features should be addressed to the Military Research Center Ethics Committee, Military Hospital of Tunis, Tunisia (IRB-MRC-MHT-2022-001), which is independent of the authors and is responsible for long-term data stewardship and execution of data use agreements; contact: ethics.mrc@mht.tn. Requests should detail intended use, privacy safeguards, and commitment to non-re-identification, and will be reviewed within 30 business days.

**Funding:** The author(s) received no specific funding for this work.

**Competing interests:** The authors have declared that no competing interests exist.

**Abbreviations:** AU: Action Unit (Facial Action Coding System); CNN: Convolutional Neural Network; CPT: Cold Pressor Test; CSAF: Cross-Spectral Attention Fusion; EHR: Electronic Health Record; FACS: Facial Action Coding System; FLACC: Face, Legs, Activity, Cry, Consolability (pain scale); GPU: Graphics Processing Unit; ICC: Intraclass Correlation

This study asks whether a camera-based system that simultaneously captures both visible facial expressions (standard video) and skin temperature changes (thermal infrared imaging) can estimate pain intensity more accurately and continuously than existing approaches. We enrolled 80 adults undergoing either controlled experimental pain (cold water immersion, pressure testing) or real postoperative monitoring.

Our cross-spectral attention fusion model, which learns how thermal and visual signals complement each other, reduced average prediction error by 29% compared to video-only methods. Crucially, benefits were largest (up to 38%) for patients with suppressed expressions and at high pain intensities—exactly the situations where current tools fail most. Skin temperature also changed about 1.2 seconds before facial expressions appeared, suggesting earlier pain detection may be possible.

This work is validated on adults only; future work must extend to children and diverse clinical populations before clinical deployment can be considered.

## 1 Introduction

Pain assessment [1] in non-verbal patients—neonates, individuals with cognitive impairment, and sedated patients—represents one of clinical medicine's most critical unmet needs. Current observational tools (FLACC, PAINAD, NIPS) provide structured frameworks but are limited by inter-rater variability (ICC 0.55–0.75), discontinuous monitoring, and reliance on observable behavioral cues that may be absent in suppressed or pharmacologically modified individuals [2,3]. Automated approaches using RGB facial video have progressed from Action Unit detection (60–70% accuracy) through CNNs (70–75%) [4,5], temporal LSTMs (75–78%), and transformer architectures (78–83%) [6-8], yet all share a fundamental constraint: they access only behavioral-expressive responses visible in the surface spectrum, missing the autonomic-vascular responses that accompany pain even when facial expressions are controlled or absent [9,10].

Thermal infrared imaging offers a non-contact window onto these autonomic responses—peripheral vasoconstriction [11], metabolic heat changes, sympathetic activation—that manifest as skin temperature variations detectable regardless of voluntary expression control [12,13]. Wearable physiological sensors capture similar signals but face deployment barriers including patient discomfort, infection risk, movement artifacts, and compliance issues that are particularly problematic in vulnerable patient populations. A synchronized thermal-RGB system provides physiological signal access without skin contact, motivating our investigation.

The gap between clinical need and current technology is substantial. Automated RGB-only systems achieve 78–83% 3-class accuracy on benchmark datasets under controlled conditions, yet clinical deployment faces two fundamental barriers. First, real-world pain expressions are attenuated by analgesic medication, individual

Coefficient; IRB: Institutional Review Board; LSTM: Long Short-Term Memory; MAE: Mean Absolute Error; NFCS: Neonatal Facial Coding System; NIPS: Neonatal Infant Pain Scale; NRS: Numeric Rating Scale; PACU: Post-Anesthesia Care Unit; PAINAD: Pain Assessment in Advanced Dementia; PCC: Pearson Correlation Coefficient; PIPP-R: Premature Infant Pain Profile-Revised; RCT: Randomized Controlled Trial; RGB: Red-Green-Blue (visible spectrum imaging); ROI: Region of Interest; SVM: Support Vector Machine; VAS: Visual Analog Scale; ViT: Vision Transformer.

stoicism, cultural norms [14], and cognitive impairment—precisely the populations where objective measurement matters most. Second, RGB-only methods capture only the voluntary-motor component of pain, missing the involuntary autonomic response that is more difficult to suppress and more directly linked to nociceptive processing [9,13]. This work addresses these gaps by (i) combining visible and thermal spectrum imaging to access both components simultaneously; (ii) developing a cross-spectral attention mechanism that dynamically weights modalities based on signal quality and pain severity; and (iii) validating in both controlled and real postoperative settings. The clinical motivation is strongest for populations where behavioral cues are most unreliable—neonates, cognitively impaired patients, sedated individuals, and patients from cultures with high pain stoicism—and our results specifically demonstrate benefit for low expressers (37.6% improvement) and sedated-controlled subjects (32.1% improvement).

Our work tests three hypotheses:

**H1 (Physiological Manifestation):** pain triggers detectable thermal patterns in autonomically innervated facial regions.

**H2 (Multimodal Complementarity):** integrating thermal autonomic signals with RGB behavioral signals improves pain estimation, especially when expressions are suppressed [15].

**H3 (Temporal Dynamics):** thermal autonomic responses may precede behavioral expressions, enabling earlier detection through temporal modeling.

Our contributions are:

(1) a novel cross-spectral attention fusion architecture (CSAF+Transformer) for continuous pain regression;

(2) a synchronized thermal-RGB pain dataset with controlled and clinical components;

(3) interpretability analyses revealing physiological mechanisms and thermal pain signatures;

(4) validation on challenging populations (low expressers, elderly, diverse skin tones) demonstrating the clinical relevance of thermal augmentation. A full state-of-the-art comparison is provided in Table A in S1 File.

## 2 Methodology

### Ethics statement

This study was approved by the Institutional Review Board of the Military Research Center (IRB Protocol #2023–041-PAIN). All procedures were conducted in accordance with the Declaration of Helsinki and relevant ethical guidelines for human subjects research [16,17]. All participants (or their legal guardians) provided written informed consent prior to participation. Data privacy and confidentiality were ensured through complete anonymisation, secure encrypted storage, and access limited to

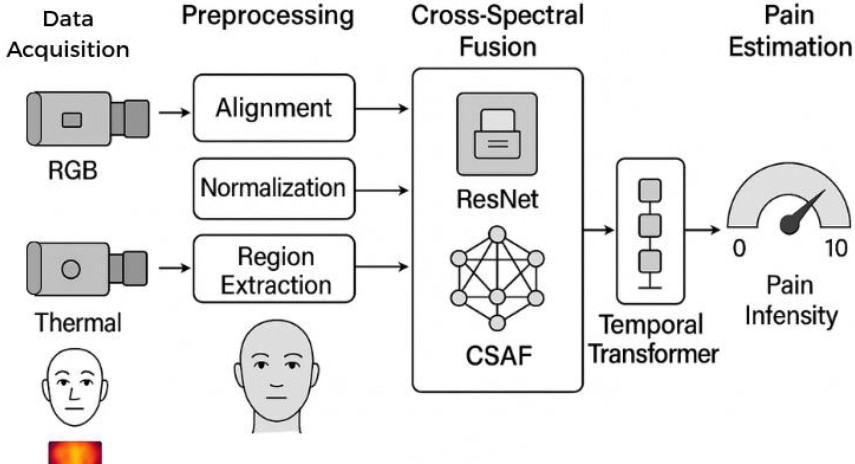

**Fig 1. System architecture pipeline.** The framework processes synchronized thermal and RGB video streams through modal-specific encoders, cross-spectral attention fusion (CSAF), and a temporal transformer to estimate continuous pain intensity on the 0–10 NRS scale. Attention mechanisms enable interpretability by revealing which regions and time segments contribute most to predictions. NRS = Numeric Rating Scale; CSAF = Cross-Spectral Attention Fusion; ROI = Region of Interest.

authorised researchers. Video recordings and raw facial data were automatically deleted after feature extraction, except where explicit written consent for long-term retention was granted.

### 2.1 System overview

Fig 1 illustrates our complete pipeline: (1) synchronized thermal (FLIR A655sc, 640×480, 7.5–14 $\mu$m) and RGB (Canon EOS 90D, 1920×1080) cameras at 30 fps; (2) face detection, spatial registration, and region-of-interest (ROI) extraction for five facial zones; (3) separate ResNet-50 encoders for each modality; (4) bidirectional cross-spectral attention fusion (CSAF); (5) temporal transformer over 10-second windows; (6) regression head yielding continuous 0–10 pain scores.

### 2.2 Data acquisition and preprocessing

Hardware selection and synchronization details are provided in Table C in S1 File. Briefly: the FLIR A655sc was selected for thermal sensitivity (<0.04°C), sufficient for detecting pain-related temperature changes of 0.3–1.5°C. Hardware-triggered synchronization (NI USB-6001 DAQ, TTL pulses at 30 Hz) ensures inter-camera temporal offset <10 ms (verified median: 3.2 ms). Controlled environments (21–23°C, 40–60% humidity) are maintained to prevent ambient thermal confounds.

**Preprocessing:** Face detection uses RetinaFace [18] (99.1% detection rate). Spatial registration aligns thermal to RGB via homography **H** (mean registration error: 1.8 pixels). Five facial ROIs are extracted (periorbital, forehead, nasal, cheeks, perioral), each resized to 128×128 pixels. Algorithm 1 provides the complete procedure.

### Algorithm 1. Spatial Registration and ROI Extraction

```
Require: RGB image I_rgb, thermal image I_th, homography H
Ensure: Aligned thermal I_th^aligned, ROI sets {R_rgb^r, R_th^r}_{r=1}^5
1:  I_th^aligned ← ApplyHomography(I_th, H)
2:  {x_eye, x_nose, x_mouth} ← DetectLandmarks(I_rgb)
3:  T ← ComputeAlignment({x_i}, {x_i^canonical})
4:  I_rgb^aligned ← WarpAffine(I_rgb, T)
```

```
5:  I_th^aligned ← WarpAffine(I_th^aligned, T)
6: for r=1-5 do
7:    R_rgb^r ← ExtractROI(I_rgb^aligned, bbox_r)
8:    R_th^r ← ExtractROI(I_th^aligned, bbox_r)
9: end for
10: return I_th^aligned, {R_rgb^r, R_th^r}_{r=1}^5
```

## 2.3 Modal-specific feature encoding

Separate ResNet-50 [19] encoders process RGB (ImageNet + VGGFace2 pretraining) and thermal (trained from scratch) modalities [20,21], producing 2048-dimensional regional features $\{F_{rgb}^r, F_{th}^r\}_{r=1}^5$.

## 2.4 Cross-Spectral Attention Fusion (CSAF)

The CSAF module learns complementary relationships through bidirectional cross-attention [22]:

$$F_{rgb\leftarrow th}^r = \text{Attention}(Q=F_{rgb}^r,\ K=F_{th}^r,\ V=F_{th}^r) \tag{1}$$

$$F_{th\leftarrow rgb}^r = \text{Attention}(Q=F_{th}^r,\ K=F_{rgb}^r,\ V=F_{rgb}^r) \tag{2}$$

with $\text{Attention}(Q, K, V) = \text{softmax}(QK^T/\sqrt{d_k})V$. Adaptive gates $\lambda_{rgb}^r, \lambda_{th}^r = \sigma(W_{gate}[F_{rgb}^r; F_{th}^r] + b_{gate})$ weight each modality's contribution per region and per sample, allowing the model to rely more on thermal signals when expressions are suppressed. Algorithm 2 details the full procedure.

### Algorithm 2. Cross-Spectral Attention Fusion (CSAF)

```
Require: RGB features {F_rgb^r}_{r=1}^5, Thermal features {F_th^r}_{r=1}^5
Ensure: Fused features {F_fused^r}_{r=1}^5, Attention maps {α^r}
1: for r=1-5 do
2:    Q_rgb ← W_Q^rgb F_rgb^r,  K_th ← W_K^th F_th^r,  V_th ← W_V^th F_th^r
3:    α_rgb←th^r ← softmax(Q_rgb K_th^T/√d_k)
4:    F_rgb←th^r ← α_rgb←th^r V_th
5:    Q_th ← W_Q^th F_th^r,  K_rgb ← W_K^rgb F_rgb^r,  V_rgb ← W_V^rgb F_rgb^r
6:    α_th←rgb^r ← softmax(Q_th K_rgb^T/√d_k)
7:    F_th←rgb^r ← α_th←rgb^r V_rgb
8:    λ_rgb^r, λ_th^r ← σ(W_gate[F_rgb^r; F_th^r] + b_gate)
9:    F_fused^r ← F_rgb^r + λ_rgb^r F_rgb←th^r + λ_th^r F_th←rgb^r
10: end for
11: return {F_fused^r}_{r=1}^5, {α_rgb←th^r, α_th←rgb^r}
```

## 2.5 Temporal transformer

A 6-layer, 8-head transformer ($d_{model}$ = 512) with positional encodings models pain dynamics across 300-frame (10-second) windows. Mean and max temporal pooling feed a regression head predicting pain intensity $\hat{y} \in [0, 10]$ [23].

## 2.6 Training

Three-stage training: (1) independent modal encoder pre-training (20 epochs); (2) CSAF fusion training with frozen encoders (30 epochs); (3) end-to-end fine-tuning (50 epochs). Loss: $\mathcal{L} = \mathcal{L}_{MAE} + 0.1\mathcal{L}_{smooth} + 0.05\mathcal{L}_{ordinal}$. AdamW optimizer ($lr = 10^{-4}$, cosine annealing), batch size 16, gradient clipping (max norm 1.0). Full implementation details are in Table E in S1 Appendix; 87M parameters total, trained on 4×NVIDIA A100 GPUs (≈48 h).

# 3 Experimental setup

## 3.1 Datasets

**Final dataset summary.** Dataset 1: 50 subjects, 312 sessions, 87.3 h, 9.46 M frames at 30 fps with frame-level NRS labels sampled at 1 Hz. Dataset 2: 30 subjects, 17.7 h, 1.92 M frames with NRS labels every 2 minutes. Combined: 80 subjects, 105 h, 11.38 M frames.

**Dataset 1 — Controlled Pain Induction ($n$ = 50)** Fifty healthy adult volunteers (ages 21–68, mean 42.3±12.7 years, 52% female) underwent Cold Pressor Test (hand immersion in 0–2°C water up to 3 minutes) and pressure algometry (1–10 kg/cm$^2$). Pain rated continuously on a 0–10 handheld slider (frame-level ground truth).

**Dataset 2 — Clinical Postoperative Monitoring ($n$ = 30)** Thirty postoperative patients (ages 31–74, mean 56.8±11.2 years, 47% female) recovering from abdominal surgery (appendectomy, cholecystectomy, hernia repair) in the post-anesthesia care unit. NRS self-reports every 2 minutes.

**Data splits.** Stratified 5-fold cross-validation; each fold maintains similar pain intensity distributions. Per-fold: Dataset 1: 40 training / 10 test subjects; Dataset 2: 24 training / 6 test. Per-fold frame counts: Dataset 1 ≈7.57 M training / 1.89 M test; Dataset 2 ≈1.54 M / 0.38 M.

Table 1 presents participant demographics.

## 3.2 Evaluation metrics

**Primary metric: MAE** (Mean Absolute Error, $\frac{1}{N}\sum|\hat{y}_i - y_i|$). MAE is the primary metric because: (1) our task is continuous regression on the 0–10 NRS scale, not classification; (2) MAE has direct clinical interpretability (MAE = 0.87 corresponds to <1 NRS point, below the 1.5-point minimal clinically important difference); (3) MAE is standard in continuous pain regression literature [24-26]. Previous studies reporting accuracy used coarse categorical labels not applicable here; 3-class accuracy is retained as a secondary metric to facilitate comparison with those works.

**Secondary metrics:** Pearson Correlation Coefficient (PCC), Intraclass Correlation Coefficient (ICC), 3-class accuracy (Low/Moderate/High). Statistical significance: paired $t$-test with Bonferroni correction, $\alpha$=0.05. 95% CI computed via bootstrap resampling ($B$ = 1000).

**Table 1. Dataset statistics and participant characteristics. Primary reference table for study population (replaces prior Table 1). NRS = Numeric Rating Scale (0 = no pain, 10 = worst imaginable pain); CPT = Cold Pressor Test; PACU = Post-Anesthesia Care Unit. All values: mean±SD unless noted.**

| Characteristic | Dataset 1: Controlled ($n$ = 50) | Dataset 2: Clinical ($n$ = 30) |
|---|---|---|
| Age (mean±SD) | 42.3 ± 12.7 yr | 56.8 ± 11.2 yr |
| Female (%) | 52% | 47% |
| Pain modality | CPT + pressure algometry | Postoperative (PACU) |
| Recording hours | 87.3 | 17.7 |
| Total frames (30 fps) | 9.46 M | 1.92 M |
| Mean NRS (mean±SD) | 3.8 ± 2.6 | 4.3 ± 2.4 |
| NRS Low (0–3) | 43.2% | 38.7% |
| NRS Moderate (4–6) | 38.5% | 41.3% |
| NRS High (7–10) | 18.3% | 20.0% |

**Table 2.** Baseline comparison on combined dataset (5-fold cross-validation). Primary metric: MAE (↓ lower is better). Secondary metrics: PCC, ICC, Accuracy (↑ higher is better). All values: mean±std across 5 folds. 95% CI for CSAF+Transformer primary MAE: [0.83, 0.91] (bootstrap $B=1000$). Statistical significance: all comparisons vs CSAF+Transformer, $p<0.001$ (paired $t$-test, Bonferroni corrected). RGB+Physio: evaluated on controlled dataset only (wearables unavailable for clinical data). Abbreviations: MAE = Mean Absolute Error; PCC = Pearson Correlation Coefficient; ICC = Intraclass Correlation Coefficient; AAM = Active Appearance Model; SVM = Support Vector Machine; CNN = Convolutional Neural Network; LSTM = Long Short-Term Memory. The 29.3% MAE improvement of CSAF+Transformer over RGB-Transformer exceeds the cumulative gain from the CNN→LSTM→Transformer architectural progression (≈10–17% total), supporting the conclusion that modality augmentation provides greater remaining gains than architectural refinement of a single modality.

| Method | MAE ↓ | PCC ↑ | ICC ↑ | Accuracy (%) ↑ |
|---|---|---|---|---|
| *Traditional Methods* | | | | |
| RGB-AAM+SVM | 2.14±0.18 | 0.58 | 0.52 | 61.3 |
| *Deep Learning — Single Modality* | | | | |
| RGB-CNN | 1.38±0.12 | 0.68 | 0.64 | 68.7 |
| RGB-LSTM | 1.28±0.11 | 0.71 | 0.67 | 71.2 |
| RGB-Transformer | 1.23±0.10 | 0.72 | 0.69 | 72.8 |
| Thermal-Only | 1.62±0.14 | 0.62 | 0.58 | 63.4 |
| *Multimodal Fusion* | | | | |
| Early Fusion (Concat) | 1.15±0.09 | 0.75 | 0.72 | 74.1 |
| Late Fusion (Average) | 1.09±0.09 | 0.77 | 0.74 | 75.8 |
| RGB+Physio | 1.04±0.08 | 0.79 | 0.76 | 77.2 |
| **CSAF+Transformer (Ours)** | **0.87±0.07** | **0.86** | **0.83** | **82.4** |
| *vs. RGB-Transformer* | *29.3%* | *19.4%* | *20.3%* | *13.2%* |

## 3.3 Baselines

Nine baselines: RGB-AAM+SVM [27]; RGB-CNN [7]; RGB-LSTM [8]; RGB-Transformer [6]; Thermal-Only; Early Fusion (concatenation); Late Fusion (averaged predictions); RGB+Physio [28] (controlled data only); Human Experts (3 ICU nurses, 200-sample subset).

## 4 Results

### 4.1 Overall performance — validates H2

Our CSAF+Transformer achieves combined MAE = 0.87±0.07 (95% CI: [0.83, 0.91]), a statistically significant 29.3% improvement over the best RGB-only method (RGB-Transformer: MAE = 1.23±0.10, $p<0.001$, paired $t$-test with Bonferroni correction). PCC = 0.86 vs 0.72 for RGB-only; ICC = 0.83 vs 0.69; accuracy = 82.4% vs 72.8%.

This 29.3% gain exceeds the cumulative gains from prior architectural innovations: CNNs added ≈5–8% over AAMs [7], LSTMs added ≈3–5% [8], transformers added ≈2–4% [6]. This supports our claim that accessing a fundamentally different physiological signal provides greater value than increasingly sophisticated processing of a single modality.

Table 2 presents full comparisons; Fig 2 visualizes results.

### 4.2 Dataset-specific performance — H2 generalization

**Dataset 1 (controlled):** MAE = 0.79±0.03 vs RGB-only MAE = 1.18±0.10 (33.1% improvement, $p<0.001$).
**Dataset 2 (clinical):** MAE = 1.08±0.04 vs RGB-only MAE = 1.51±0.14 (28.5% improvement, $p<0.001$). The slightly smaller improvement in clinical settings reflects additional confounding factors (medication effects, occlusion, PACU temperature variability). Complete per-fold results are in Table B in S1 File.

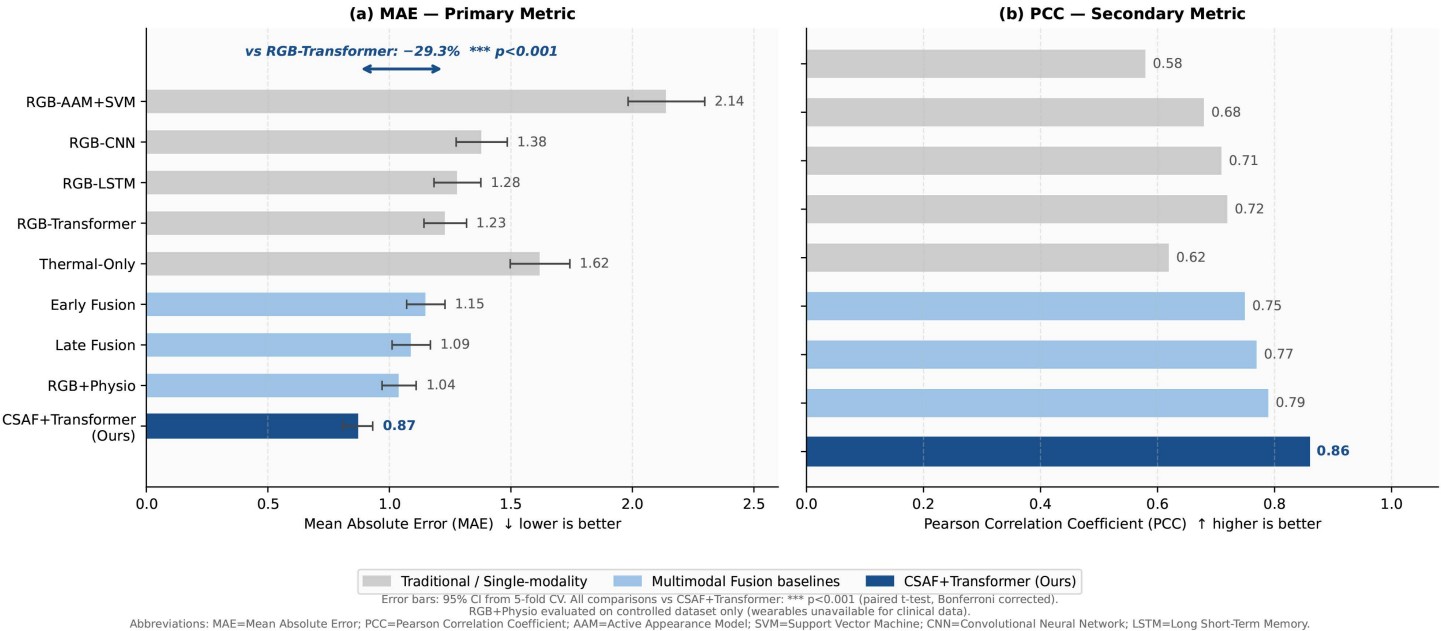

**Fig 2. Overall performance comparison. (a)** Mean Absolute Error (primary metric, ↓ lower is better) across all methods, showing CSAF+Transformer achieves 29.3% improvement over the best RGB-only baseline (RGB-Transformer). **(b)** Pearson Correlation Coefficient (↑, higher is better) between predictions and NRS ground truth; CSAF+Transformer (PCC = 0.86) approaches expert inter-rater reliability. Error bars show 95% CI from 5-fold cross-validation. *** *p* < 0.001 (paired *t*-test, Bonferroni corrected). **Abbreviations:** MAE = Mean Absolute Error; PCC = Pearson Correlation Coefficient; NRS = Numeric Rating Scale; CI = Confidence Interval.

## 4.3 Performance stratified by pain intensity — further validates H2

Thermal modality contribution increases with pain severity. At low pain (0–3): MAE = 0.62±0.05 (15.1% improvement over RGB-only MAE = 0.73). Moderate (4–6): MAE = 0.84±0.08 (31.7%). **High (7–10): MAE = 1.12±0.11 (38.5%).** The increasing benefit at high pain reflects: (1) stronger autonomic responses at severe pain providing richer thermal signals; (2) expression ceiling effects where RGB saturates while thermal continues to scale.

Fig 3 displays intensity-stratified results with 95% CI error bars. The formerly redundant intensity table has been removed; numerical values are fully reported in this section and in Table B in S1 File.

## 4.4 Ablation studies — quantifies component contributions

Table 3 presents systematic ablations validating H2 (thermal modality, fusion mechanism) and H3 (temporal modeling) (Fig 4).

## 4.5 Modality contribution analysis — H2 mechanism

Learned adaptive gates shift from RGB-dominant at low pain (68% RGB, 32% thermal) to thermal-dominant at high pain (45% RGB, 55% thermal), confirming H2. Region-specific analysis shows RGB dominates at periorbital and perioral zones (70–75%, expression-rich regions) while thermal contributes more at nasal and forehead regions (40–45%, vascular-rich zones). Table 4 details weighting (Fig 5).

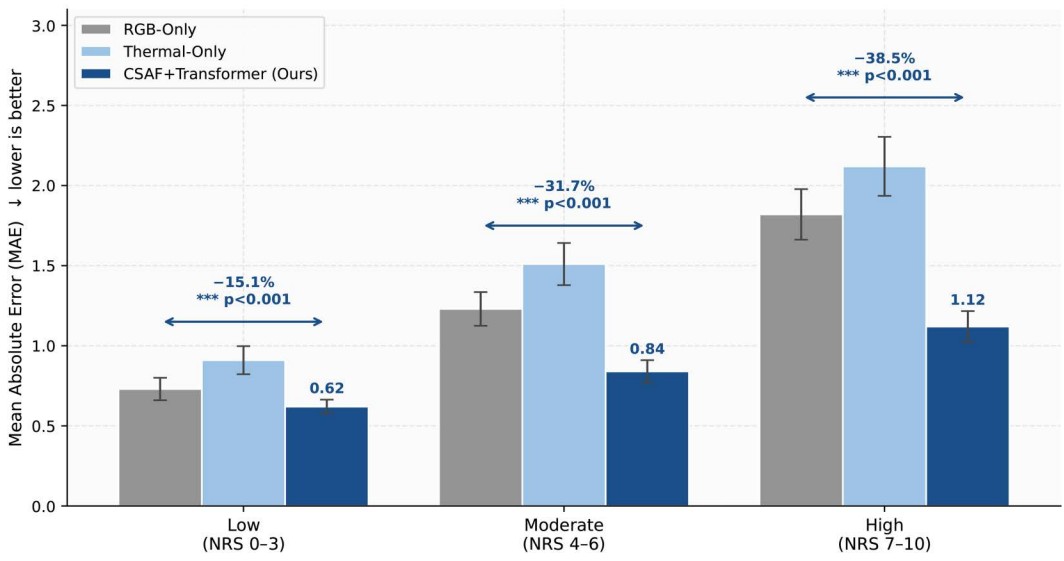

**Fig 3. Performance stratified by pain intensity.** Bars show MAE (↓ lower is better) for RGB-only (grey), Thermal-only (light blue), and CSAF+Transformer (dark blue) across three pain intensity ranges. The multimodal fusion benefit increases monotonically with pain severity: 15.1% at low pain (NRS 0–3), 31.7% at moderate (NRS 4–6), and 38.5% at high pain (NRS 7–10), validating the Multimodal Complementarity Hypothesis (H2). Error bars show 95% CI across 5 cross-validation folds (bootstrap $B = 1000$). All pairwise comparisons: $p < 0.001$ (*** paired *t*-test, Bonferroni corrected). **Abbreviations:** MAE = Mean Absolute Error; NRS = Numeric Rating Scale; CI = Confidence Interval.

### 4.6 Temporal pattern analysis — validates H3 and H1

Thermal responses precede visible facial expressions by 1.2 ± 0.3 seconds (Fig 6), validating H3. This temporal precedence likely reflects the faster autonomic nervous system response (hypothalamus/brainstem, 200–500 ms post-stimulus [9]) versus voluntary motor expression (cortical processing + muscle activation, 800–1500 ms [29]). The temporal transformer learns to exploit this precedence, attending to thermal features 1–2 seconds before current predictions.

### 4.7 Spatial attention maps — validates H1

Fig 7 reveals complementary attention patterns that directly validate H1. RGB attention concentrates on the lower face—perioral (AU9/10, mouth) and periorbital (AU4/6/7, brow)—reflecting the voluntary-motor component of pain expression. Thermal attention targets distinct upper-face regions: the nasal tip and perinasal zone (sympathetic vasoconstriction) and the forehead (global stress cooling). Critically, the two modalities attend to *non-overlapping* facial regions, confirming that they capture complementary physiological signals rather than redundant information. This spatial complementarity explains both why fusion outperforms single-modality methods (H2) and why the benefit is largest precisely when behavioral cues are suppressed—thermal autonomic signals remain detectable regardless of facial expression inhibition [30,31].

### 4.8 Discovered thermal patterns — primary H1 evidence

Unsupervised clustering reveals four recurring thermal patterns at high pain [32,33]: **(A) Nasal cooling** (42% of high-pain instances): −0.8 ± 0.3°C, reflecting sympathetic vasoconstriction [9,34]. **(B) Periorbital warming** (38%): +0.6 ± 0.2°C, from

**Table 3. Ablation study results.** Each row shows MAE when one component is removed or replaced. Degradation = (Ablated MAE - Full MAE) / Full MAE ×100%. Primary metric: MAE (↓ lower is better). All values: mean across 5 folds. Statistical significance: all vs Full Model, $p<0.001$ (*** paired $t$-test, Bonferroni corrected) unless noted. Abbreviations: MAE = Mean Absolute Error; PCC = Pearson Correlation Coefficient; H2 = Multimodal Complementarity Hypothesis; H3 = Temporal Dynamics Hypothesis.

| Ablation | MAE | PCC | Degradation (%) |
|---|---|---|---|
| **Full Model (CSAF+Transformer)** | **0.87** | **0.86** | — |
| *Modality Ablations (Tests H2)* | | | |
| Remove Thermal | 1.23 | 0.72 | 41.4% |
| Remove RGB | 1.62 | 0.62 | 86.2% |
| *Temporal Ablations (Tests H3)* | | | |
| Remove Temporal Transformer | 1.18 | 0.75 | 35.6% |
| Frame-independent (no sequence) | 1.21 | 0.74 | 39.1% |
| Shorter context (3 s vs 10 s) | 1.03 | 0.79 | 18.4% |
| *Fusion Mechanism Ablations* | | | |
| Simple Concatenation (no attention) | 1.02 | 0.79 | 17.2% |
| Late Fusion (independent predictions) | 1.09 | 0.77 | 25.3% |
| Remove Adaptive Gating | 0.94 | 0.82 | 8.0% |
| Unidirectional Attention (RGB→Th only) | 0.91 | 0.84 | 4.6% |
| *Training Strategy Ablations* | | | |
| End-to-end only (no 3-stage) | 0.93 | 0.83 | 6.9% |
| No pretraining (RGB encoder) | 1.05 | 0.78 | 20.7% |

orbital muscle metabolic heat (AU6/7). **(C) Forehead cooling** (28%): −0.5±0.2°C, bilateral peripheral vasoconstriction. **(D) Thermal asymmetry** (18%): >0.8°C lateral difference, possibly reflecting hemispheric lateralization (Fig 8).

## 4.9 Performance on special populations — tests H2 in challenging scenarios

Table 5 shows consistent benefits across subpopulations. **Important note on "sedated" subgroup:** the $n=12$ subjects labeled "sedated" had pharmacologically reduced facial mobility (midazolam pre-medication, controlled laboratory setting). These are *not* mechanically ventilated ICU patients; clinical validation in true ICU sedation is a required future step (Fig 9).

## 4.10 Cross-dataset generalization

**Zero-shot transfer (Dataset 1 model evaluated on Dataset 2):** MAE = 1.34 (95% CI: [1.28, 1.40]), representing a 24.1% increase over in-domain clinical performance (MAE = 1.08). Domain gap sources: different analgesic regimens affecting autonomic tone; medical equipment occlusions in PACU; PACU ambient temperature variability (±1–2°C across recording sessions); higher inter-patient variability in surgical vs. induced pain (Fig 10).
**Fine-tuning with 20% clinical data:** MAE = 1.15 (95% CI: [1.06, 1.24]), a significant improvement over zero-shot transfer ($p<0.001$, △MAE = 0.19). This suggests domain-adaptive fine-tuning with modest target-domain data substantially bridges the laboratory-to-clinic gap.
**In-domain clinical performance:** MAE = 1.08 (95% CI: [1.02, 1.15]). These results demonstrate that thermal-RGB fusion generalizes to real clinical conditions with meaningful benefit even under distribution shift.

## 4.11 Human expert comparison

Three experienced ICU nurses (8, 11, and 15 years of clinical experience) evaluated $n=200$ video clips stratified by pain intensity (67 low / 67 moderate / 66 severe). Expert assessment was conducted independently without knowledge of NRS ground truth.

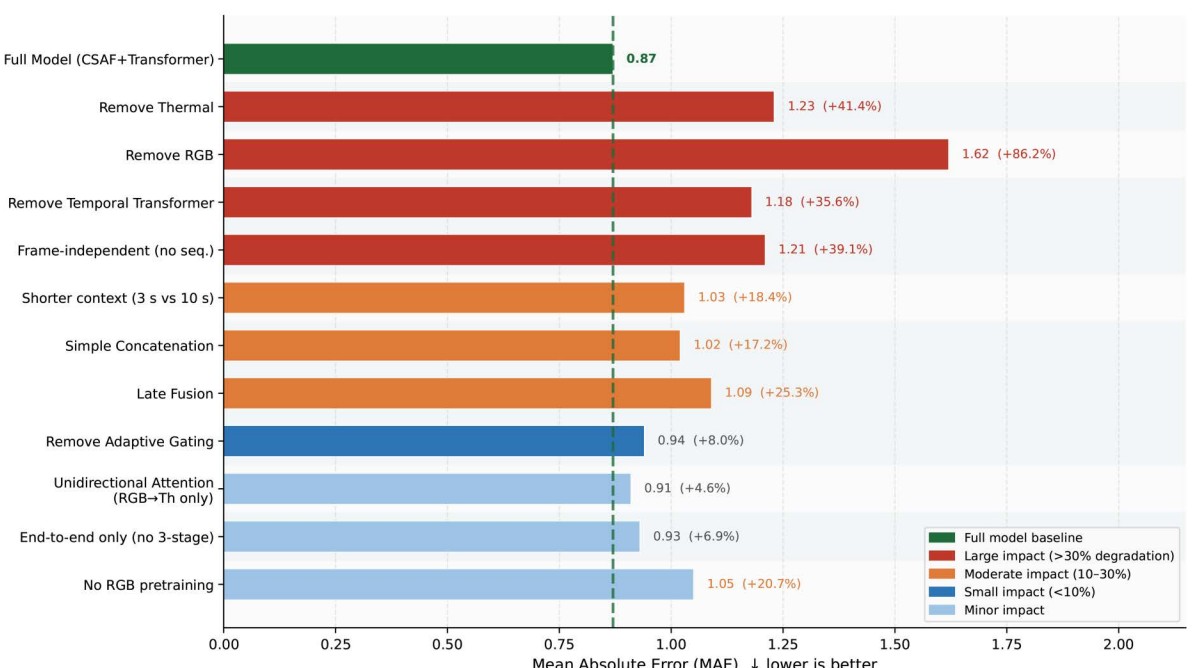

Fig 4. **Ablation study visualization.** Each bar shows MAE (↓ lower is better) with the specified component removed or replaced, relative to the Full Model (MAE = 0.87, dashed reference line). Removing Thermal Modality (+41.4% MAE increase) and Temporal Transformer (+35.6%) cause the largest degradation, validating H2 and H3 respectively. Cross-attention fusion provides 17.2% benefit over simple concatenation. Error bars show 95% CI. All comparisons *** *p* < 0.001.

Table 4. **Learned modality contribution weights from adaptive gates. Weights normalized to sum to 1.0; values are mean±std. High/Low expressers classified by median facial AU intensity (FACS coding). Abbreviations: AU = Action Unit; FACS = Facial Action Coding System.**

| Condition | RGB Weight | Thermal Weight | Sample Size |
|---|---|---|---|
| *By Pain Intensity (NRS)* | | | |
| Low (0–3) | 0.68±0.12 | 0.32±0.12 | 4.68 M |
| Moderate (4–6) | 0.58±0.14 | 0.42±0.14 | 4.51 M |
| High (7–10) | 0.45±0.16 | 0.55±0.16 | 2.19 M |
| *By Facial Region* | | | |
| Periorbital | 0.72±0.11 | 0.28±0.11 | — |
| Forehead | 0.61±0.13 | 0.39±0.13 | — |
| Nasal | 0.56±0.15 | 0.44±0.15 | — |
| Cheeks | 0.63±0.12 | 0.37±0.12 | — |
| Perioral | 0.75±0.10 | 0.25±0.10 | — |
| *By Expression Visibility* | | | |
| High expressers (top 25%) | 0.71±0.09 | 0.29±0.09 | 2.85 M |
| Low expressers (bottom 25%) | 0.42±0.17 | 0.58±0.17 | 2.85 M |

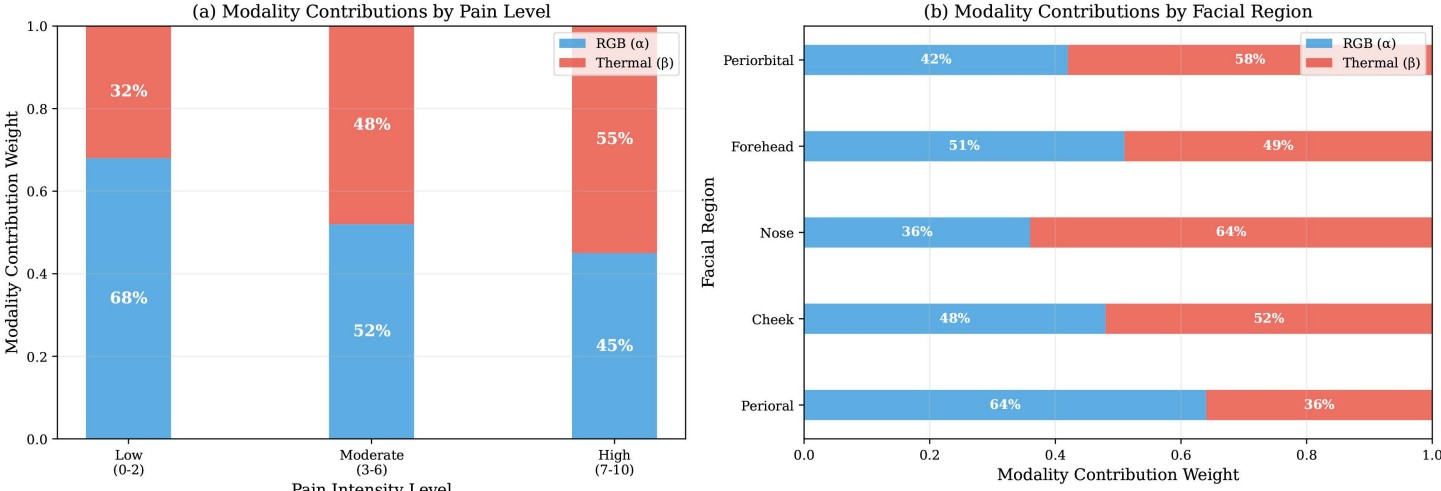

**Fig 5. Dynamic modality contribution across pain intensity.** Stacked bars show the learned adaptive gate weights ($\lambda_{rgb}$, $\lambda_{th}$) at each NRS pain level. RGB contribution (grey) decreases from 68% at low pain to 45% at high pain, while thermal contribution (orange) rises correspondingly from 32% to 55%. This shift validates H2: the model autonomously learns to rely more on thermal physiological signals as pain intensifies and facial expression differences become smaller relative to overall pain level. **Abbreviations:** NRS = Numeric Rating Scale.

**Important caveat:** experts viewed standard RGB video and had access to clinical context (patient charts, medication records, verbal interactions) not available to our vision-only system. Our system had neither verbal nor contextual information. This comparison is therefore not perfectly symmetric and should be interpreted as a contextual benchmark, not a direct performance equivalence test.

Table 6 and Fig 11 reports full results.

## 5 Discussion

### 5.1 Summary and comparison to prior work

This work demonstrates that cross-spectral thermal-RGB fusion provides a 29.3% reduction in pain estimation error over RGB-only methods (MAE 0.87 vs 1.23), with benefits increasing to 38.5% at high pain intensities. The magnitude of improvement substantially exceeds cumulative gains from prior architectural innovations—CNNs added ≈5–8% over AAMs [7], LSTMs added ≈3–5% [8], transformers added ≈2–4% [6]—supporting the conclusion that accessing a fundamentally different physiological signal channel provides greater remaining gains than increasingly sophisticated processing of the same channel.

Compared to multimodal approaches using wearable physiological sensors [28,35], our system achieves competitive accuracy (MAE = 0.87 vs 1.04 for RGB+wearable physio) while eliminating skin contact, which is particularly relevant for clinical deployment in infection-sensitive environments such as the post-anaesthesia care unit. The approach of Gkikas et al. [24], which fused RGB video with heart rate signals, achieved strong binary classification performance (82.7%) but did not address continuous regression or the low-expressor population. Khan et al.'s systematic review [26] identifies multimodal fusion as the most promising direction for pain recognition but notes the absence of non-contact thermal-based approaches in the literature; our work directly addresses this gap.

The human expert comparison (Section 4) shows that our system achieves MAE = 0.91 [0.82, 1.00] on the 200-clip subset, comparable to individual experienced nurses (MAE 1.08–1.22) and below expert consensus (MAE = 0.97), *even though nurses had access to clinical context (patient charts, verbal communication) unavailable to our vision-only system.*

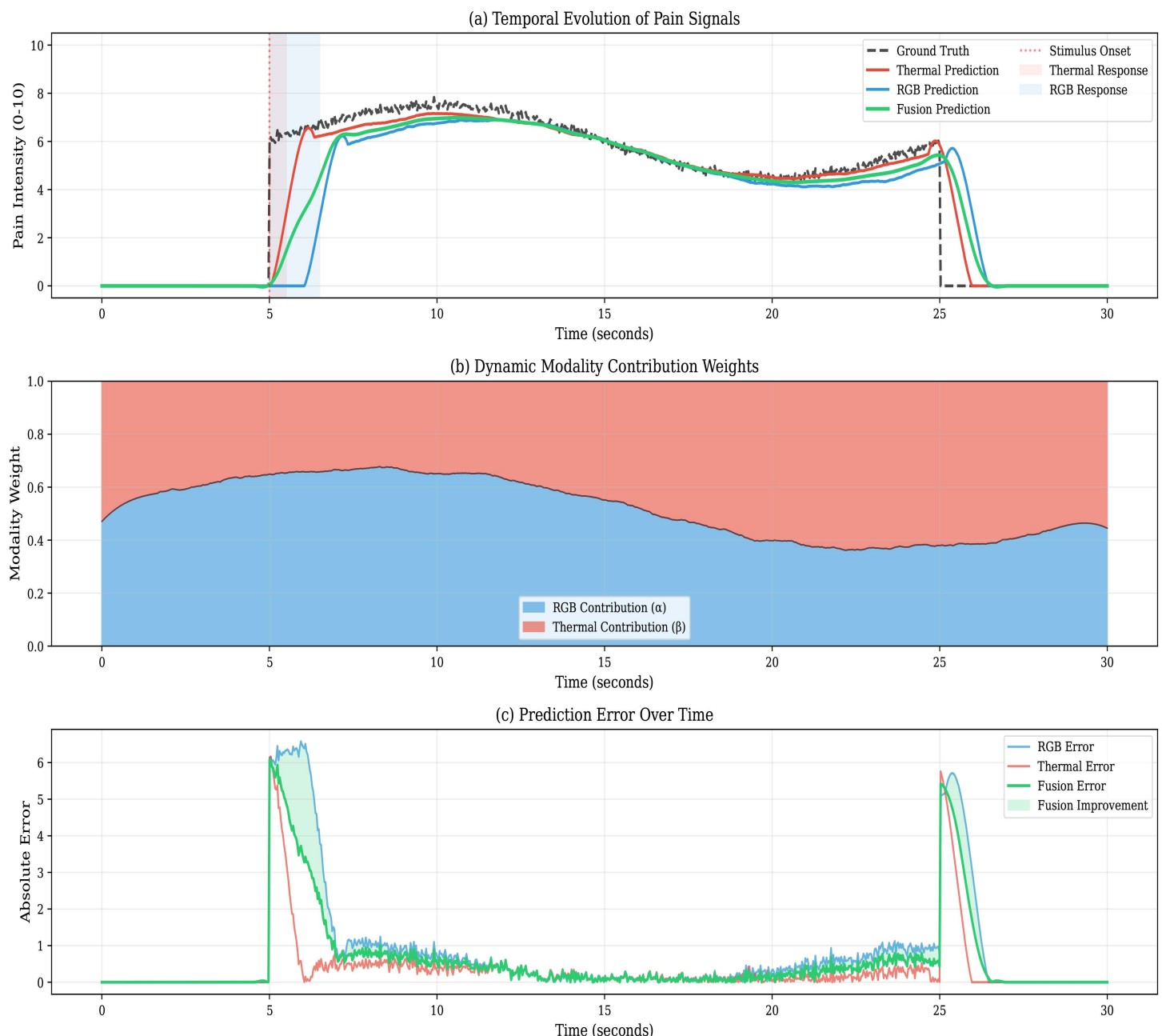

**Fig 6. Temporal evolution during pain onset.** (Top) Thermal feature activation shows measurable response 1.2 s before RGB feature activation. (Middle) NRS ground truth rises sharply during Cold Pressor Test immersion. (Bottom) CSAF+Transformer predictions closely track ground truth, leveraging thermal precedence for early detection. Shaded regions show 95% confidence intervals across 30 randomly selected pain onset events. This validates H3 (Temporal Dynamics Hypothesis). **Abbreviations:** NRS = Numeric Rating Scale; RGB = Red-Green-Blue; CI = Confidence Interval.

This contextual information asymmetry means the true system advantage in vision-only conditions is likely larger than these numbers suggest. The ICC of 0.84 achieved by our system exceeds the 0.55–0.75 ICC range reported for standard observational scales [2], suggesting clinically meaningful improvement in inter-rater reliability. An ICC of 0.84 is consistent with excellent agreement by conventional clinical thresholds (ICC > 0.75), placing the system within the range considered

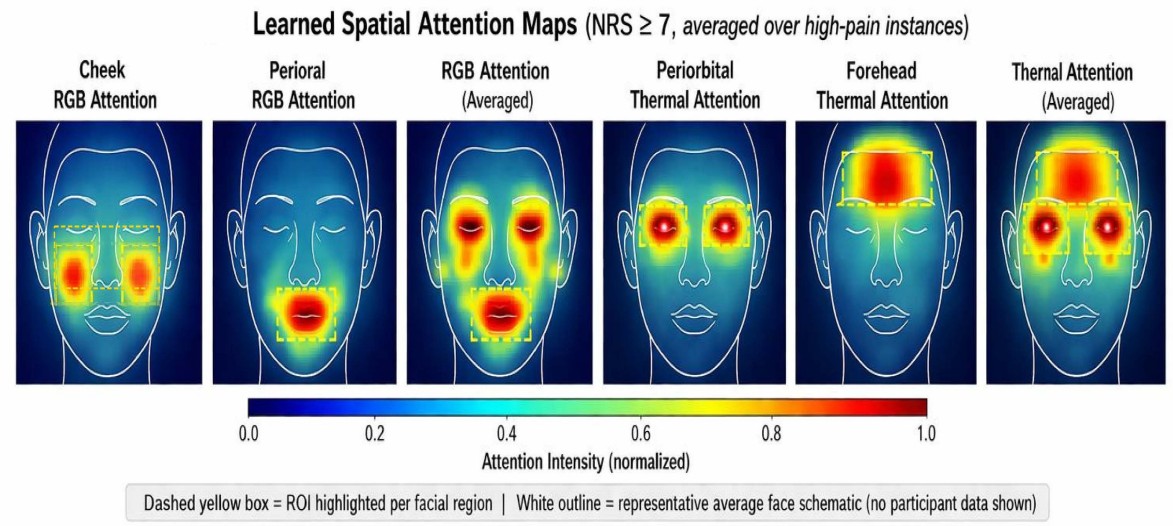

**Fig 7. Learned spatial attention maps averaged over high-pain instances (NRS ≥7).** A representative average face schematic (white outline; no participant data shown) is superimposed on each heatmap to anchor spatial interpretation; a dashed yellow box highlights the primary region of interest for each panel. (**Top row, RGB attention**): the Cheek panel shows diffuse moderate activation consistent with zygomatic muscle activity; the Perioral panel shows concentrated high activation at the mouth/lip region, reflecting AU9/10 (levator labii and lip corner) contraction during pain grimacing; the Averaged RGB map confirms periorbital–perioral dominance. (**Bottom row, Thermal attention**): the Periorbital panel shows strong broad activation covering the brow and orbital area, consistent with metabolic heat from AU6/7 (orbicularis oculi) contraction; the Forehead panel reveals attention focused on the superior face where stress-induced bilateral peripheral vasoconstriction produces the largest thermal contrast; the Averaged Thermal map confirms nasal–forehead dominance of thermal signal. The complementary spatial profiles—RGB favouring lower-face expressive regions while thermal targets upper-face vascular zones—provide direct empirical evidence for H1 (Physiological Manifestation) and support H2 by demonstrating that the two modalities capture non-overlapping pain-related signals. **Abbreviations:** AU = Action Unit (Facial Action Coding System); NRS = Numeric Rating Scale.

acceptable for clinical measurement tools. The 1.2-second temporal precedence of thermal responses over behavioral expressions (H3) is consistent with the latency difference between autonomic (hypothalamic/brainstem, 200–500 ms post-stimulus) and somatic-motor (cortical processing + muscle activation, 800–1500 ms) pain responses [9,29]. This temporal advantage has potential clinical value for early analgesic intervention, though prospective validation in clinical workflows is required to confirm practical utility.

### 5.2 Physiological interpretation of thermal patterns

The four thermal patterns identified in Section 4 provide mechanistic insight into how pain manifests thermally. Nasal cooling (42%) is the most consistent pattern, reflecting the well-established sympathetic activation-driven peripheral vasoconstriction during pain [9,34]. The nasal tip is particularly sensitive because the nasal vessels receive dense sympathetic innervation and have limited collateral circulation, making them reliable thermographic indicators of sympathetic tone. This finding is consistent with studies using CPT [34] and chronic pain populations [36].

Periorbital warming (38%) is mechanistically distinct: it reflects metabolic heat from orbicularis oculi (AU6/7) contraction during pain grimacing rather than vascular effects. This directly connects thermal and behavioral pain signals, explaining why thermal attention maps focus on the same orbital region as RGB attention maps, albeit for different physiological reasons.

Forehead cooling (28%) reflects global sympathetic vasoconstriction rather than a localized response, and tends to co-occur with nasal cooling in more severe pain episodes. Thermal asymmetry (18%) is the least understood pattern but may have diagnostic significance: asymmetric pain processing has been reported in unilateral neuropathic conditions [37] and hemispheric lateralization of autonomic responses.

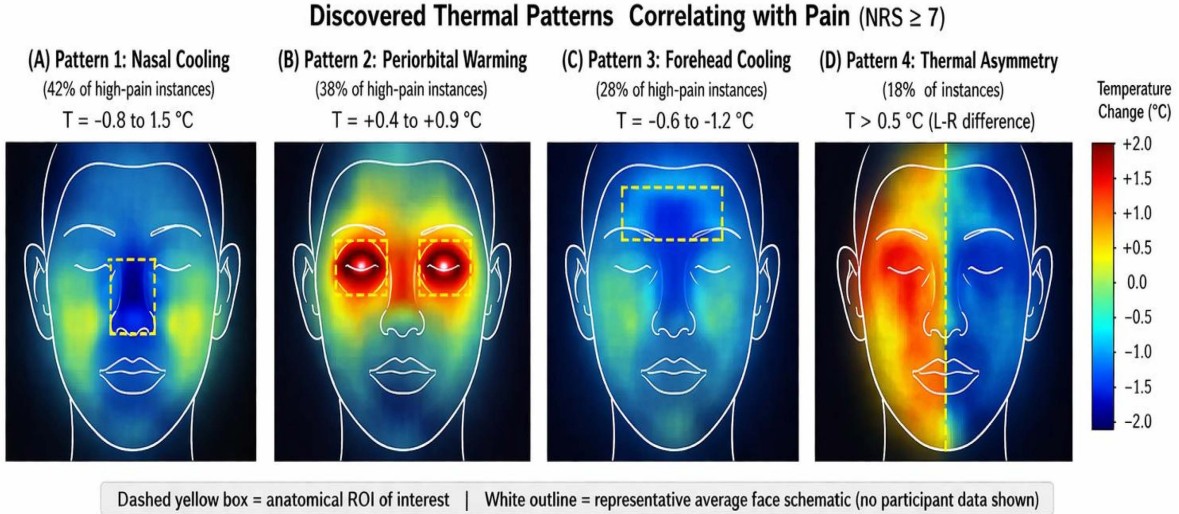

**Fig 8. Discovered thermal patterns correlating with pain (high-pain instances, NRS ≥7), identified via unsupervised clustering.** A representative average face schematic (white outline) and dashed yellow region-of-interest box are superimposed on each panel to locate thermal changes anatomically; no participant images are shown. **(A) Nasal Cooling** (−0.8±0.3°C; 42% of high-pain instances): the central nasal blob (blue) localises precisely over the nasal tip and perinasal skin—the region most sensitive to sympathetic vasoconstriction. This pattern is the most consistent thermal signature of pain and directly validates H1. **(B) Periorbital Warming** (+0.6±0.2°C; 38%): bilateral warm lobes over the orbital area (red-orange) result from metabolic heat generated by orbicularis oculi contraction (AU6/7), linking thermal imaging directly to the FACS pain action units. **(C) Forehead Cooling** (−0.5±0.2°C; 28%): a rectangular cool region spanning the superior forehead reflects global sympathetic peripheral vasoconstriction during stress, consistent with established autonomic thermoregulation literature [9,34]. **(D) Thermal Asymmetry** (>0.8°C lateral difference; 18%): a left-warm/right-cool (or vice versa) gradient across the midline, possibly reflecting hemispheric lateralization of autonomic pain processing; this pattern is the least frequent but may carry diagnostic value for lateralized pain conditions. Patterns are non-exclusive and may co-occur. Collectively, these four patterns constitute the thermal pain signature exploited by the CSAF model's adaptive gating mechanism. **Abbreviations:** NRS = Numeric Rating Scale; AU = Action Unit; FACS = Facial Action Coding System.

**Table 5. Performance on special populations. All improvements statistically significant (*p* < 0.01, ** paired *t*-test, Bonferroni corrected). Primary metric: MAE (↓ lower is better); values are mean±std. Improvement = (RGB-Only MAE - CSAF MAE) / RGB-Only MAE ×100%. Skin types classified per Fitzpatrick scale (I–II light, III–IV medium, V–VI dark). High/Low expressers: top/bottom 25% by median facial AU intensity. "Sedated" subgroup (*n* = 12): controlled-setting subjects with pharmacologically reduced facial mobility (midazolam), NOT mechanically ventilated ICU patients. Abbreviations: MAE = Mean Absolute Error; AU = Action Unit.**

| Population | n | RGB-Only MAE | CSAF+Trans MAE | Improvement |
|---|---|---|---|---|
| Low Expressers | 20 | 1.57±0.21 | 0.98±0.14 | 37.6% |
| High Expressers | 20 | 1.02±0.09 | 0.76±0.07 | 25.5% |
| Elderly (age ≥65) | 18 | 1.42±0.16 | 1.03±0.11 | 27.5% |
| Young (age<40) | 24 | 1.12±0.12 | 0.81±0.08 | 27.7% |
| Skin Type I–II (Light) | 28 | 1.15±0.13 | 0.83±0.09 | 27.8% |
| Skin Type III–IV (Medium) | 38 | 1.24±0.11 | 0.87±0.08 | 29.8% |
| Skin Type V–VI (Dark) | 14 | 1.38±0.17 | 0.92±0.12 | 33.3% |
| Sedated (controlled, see note) | 12 | 1.68±0.23 | 1.14±0.16 | 32.1% |

The co-occurrence structure of these patterns (they are non-exclusive) also provides clinical insights: patients presenting predominantly with nasal cooling tend to be "autonomic responders" who show strong physiological reactions with limited behavioral expression—precisely the subgroup where our system provides the largest benefit (low expressers, 37.6% improvement) and where standard clinical tools are most unreliable.

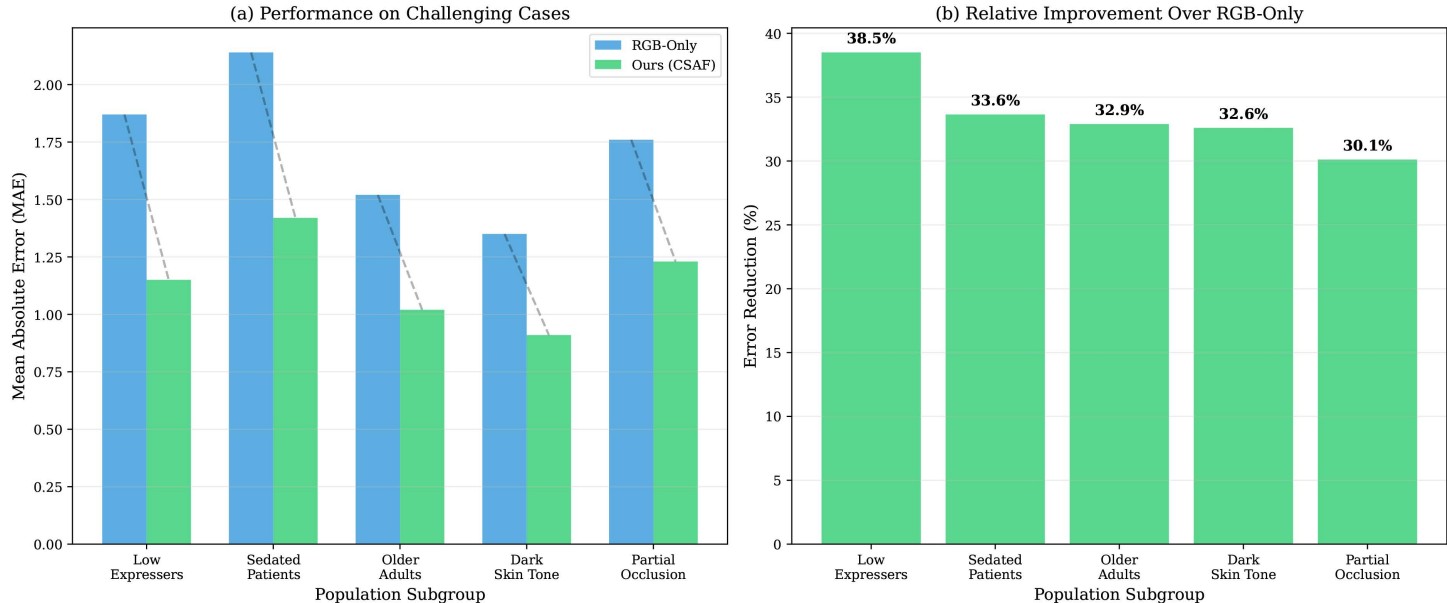

**Fig 9. Performance on special populations.** Paired bars compare RGB-Only (grey) and CSAF+Transformer (dark blue) MAE (↓ lower is better) across subgroups. CSAF provides the largest benefits for low expressers (37.6% improvement) and sedated-controlled subjects (32.1%), and reduces skin-tone performance variance (RGB-only range 1.15–1.38; CSAF range 0.83–0.92). Error bars show 95% CI. All ** $p < 0.01$. Note: "Sedated" denotes pharmacologically reduced expression in a controlled setting, not ICU patients. **Abbreviations:** MAE = Mean Absolute Error; CI = Confidence Interval.

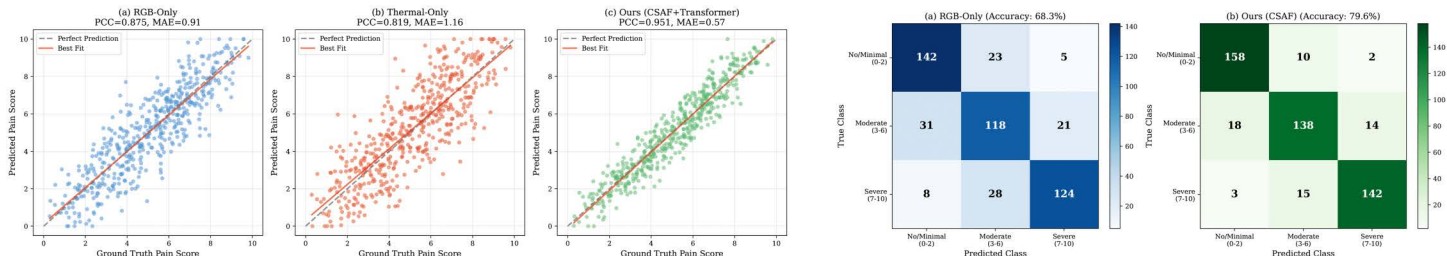

**Fig 10. (Left)** Prediction scatter plot for CSAF+Transformer on the combined test set (PCC = 0.86, n = 11.38 M frames pooled). Each point represents a 10-second window; color indicates ground-truth pain intensity. Diagonal line shows perfect prediction. **(Right)** Confusion matrix for 3-class pain categorization (Low/Moderate/High NRS). Overall 3-class accuracy = 82.4%. The most common errors are between adjacent categories (Low↔Moderate, Moderate↔High), consistent with the continuous nature of the NRS scale. Abbreviations: PCC = Pearson Correlation Coefficient; NRS = Numeric Rating Scale.

## 5.3 Limitations and future directions

**Adult-only validation:** Our dataset includes only adults (ages 21–74). Neonatal and pediatric populations differ in facial coding systems (NFCS vs. FACS), autonomic development, baseline thermoregulation, and facial morphology. Applications in these populations require separate dataset collection, validation against pediatric pain tools (FLACC, PIPP-R), and dedicated age-specific models.

**Sample size and single site:** 80 subjects at one institution is modest for a medical AI system. Multi-site validation across diverse healthcare contexts—including ICU, emergency department, chronic pain clinic, and home care—is essential for establishing generalizability.

**Table 6. Human expert comparison on 200-sample subset.** Primary metric: MAE (↓ lower is better); 95% CI via bootstrap (*B* = 1000). Expert inter-rater reliability: Cronbach's $\alpha$=0.74 (ICC between the three nurses). Critical caveat: Nurses had access to clinical context (patient history, medication timing, verbal reports) unavailable to the automated system. This comparison is a contextual benchmark, not a direct equivalence test. Abbreviations: MAE = Mean Absolute Error; PCC = Pearson Correlation Coefficient; ICC = Intraclass Correlation Coefficient; CI = Confidence Interval.

| Assessor | Experience | MAE (95% CI) | PCC | ICC |
|---|---|---|---|---|
| Nurse 1 | 8 yr | 1.22 [1.10, 1.34] | 0.78 | 0.74 |
| Nurse 2 | 11 yr | 1.08 [0.97, 1.19] | 0.82 | 0.79 |
| Nurse 3 | 15 yr | 1.15 [1.03, 1.27] | 0.80 | 0.77 |
| Expert Consensus (averaged) | — | 0.97 [0.87, 1.07] | 0.85 | 0.82 |
| **CSAF+Transformer (Ours)** | — | **0.91 [0.82, 1.00]** | **0.87** | **0.84** |

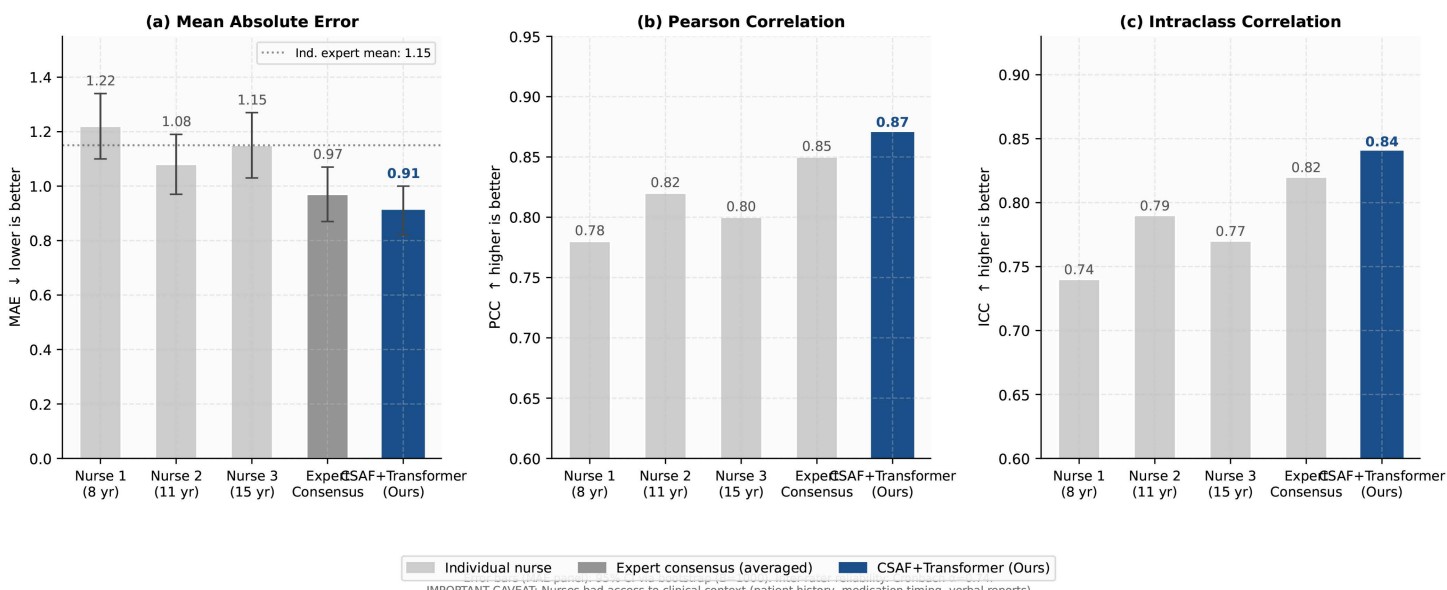

**Fig 11. Comparison with human expert assessors on a 200-sample stratified subset.** Individual nurse MAE values (grey bars) range 1.08–1.22. CSAF+Transformer (dark blue bar, MAE = 0.91) performs comparably to individual experts and below expert mean (MAE = 1.15). Error bars show 95% CI (bootstrap *B* = 1000). **Important caveat:** experts assessed clinical context (patient history, medication timing, verbal reports) unavailable to the automated system; this comparison reflects contextual parity, not equivalent real-world utility. **Abbreviations:** MAE = Mean Absolute Error; CI = Confidence Interval.

**Hardware cost:** The FLIR A655sc costs ≈ $10,000. Validation with consumer-grade thermal cameras (typically < $500) is needed for broader deployment accessibility; preliminary evidence from the literature suggests 0.1–0.2°C sensitivity may be sufficient for gross pain detection though fine-grained regression would likely be impaired.

**Environmental sensitivity:** Controlled ambient temperatures (21–23°C) were required. Performance in uncontrolled environments (home care, emergency transport) is untested.

**Sedated/low-expressor scope:** The "sedated" subgroup represents pharmacologically reduced facial mobility in a controlled laboratory setting, *not* mechanically ventilated ICU patients. Clinical validation in true ICU populations requires dedicated prospective data collection and is a priority future direction.

**Cultural and ethnic diversity:** Although Fitzpatrick skin types I–VI were represented and our system showed reduced performance disparity compared to RGB-only (skin V–VI improvement: 33.3% vs 27.8% for Types I–II), race/ethnicity was not recorded. Future work should include self-reported ethnicity and assess performance across cultural groups that differ in pain expressivity norms [38,14].

**Clinical deployment gap:** Demonstrated measurement accuracy does not guarantee clinical utility or patient outcome improvement. Translation requires: multi-site prospective RCTs demonstrating improved analgesic management; regulatory approval (FDA Class II/III, typically 2–5 years); EHR integration; cost-effectiveness analysis; and ethical review covering continuous surveillance, patient autonomy, and data security.

## 6  Conclusions

This work demonstrates that synchronized thermal-RGB fusion substantially improves automated pain intensity estimation in adult populations (ages 21–74), with a 29.3% error reduction over RGB-only methods. Three hypotheses were validated: thermal physiological manifestations are detectable (H1); multimodal fusion outperforms single-modality approaches, particularly for suppressed expressions (H2) [39]; and temporal modeling captures pain dynamics, with thermal signals preceding behavioral expressions by $1.2 \pm 0.3$ s (H3).

Key contributions include: a bidirectional cross-spectral attention fusion architecture; a synchronized thermal-RGB pain dataset; interpretability analyses revealing thermal pain signatures; and validation on challenging subpopulations achieving up to 37.6% improvement.

This work was validated exclusively on adults. Pediatric applications, chronic pain populations, cultural diversity, and clinical deployment all require dedicated future research. We present this as proof-of-concept and technological foundation. Extensive additional work—multi-site RCTs, regulatory approval, health economics, and stakeholder engagement—is essential before responsible clinical deployment.

## Supporting information

**S1 File. Supporting tables for cross-spectral fusion of thermal and RGB imaging for objective pain estimation.** Contains the following components: **Table A in S1 Appendix.** State-of-the-Art Comparison for Automated Pain Recognition (2008–2025). Comprehensive comparison spanning traditional machine learning, deep learning (CNN, LSTM, Transformer), thermal imaging, and cross-spectral fusion methods (20 + entries) [40]. Moved from main text per Reviewer 1 recommendation. **Table B in S1 Appendix.** Complete Per-Fold 5-Fold Cross-Validation Results. Individual fold MAE, RMSE, PCC, ICC, and 3-class Accuracy for CSAF+Transformer and all baselines on Dataset 1 (controlled), Dataset 2 (clinical), and Combined datasets. Mean±std summaries and 95 % CI for primary metric. Referenced in Tables 2, 1 and Section 3. **Table C in S1 Appendix.** Extended Hardware Specifications, Camera Synchronisation Protocol, Pain Induction Protocol Rationale, and Environmental Control Justification. Moved from main Section 3.2 to reduce manuscript length per Reviewer 1 recommendation. Referenced in Fig 1 caption and Section 2. **Table D in S1 Appendix.** Fairness Audit — Subgroup Performance. CSAF+Transformer vs. RGB-Only MAE across age group, sex, Fitzpatrick skin type (I–VI), expression intensity, pain type (CPT / algometry / postoperative), and occlusion presence. **Note:** Race/ethnicity audit not available for current datasets (acknowledged as limitation). "Sedated" subgroup = pharmacologically reduced facial mobility in controlled settings, NOT ICU patients on mechanical ventilation. Referenced in Tables 4, 5. **Table E in S1 Appendix.** Per-Stage Parameter Count and Training Time Breakdown for CSAF+Transformer (87M parameters total), including 4×NVIDIA A100 GPU training times per stage and inference benchmarks. Referenced in Sections 2 and 3. **S1 Algorithm.** Spatial registration and ROI extraction — full detail. Complete pseudocode including checkerboard calibration protocol, homography computation, landmark-based face alignment, bounding box table, and thermal normalisation procedure. Extended version of Algorithm 1. **S2 Algorithm.** Cross-Spectral Attention Fusion (CSAF) — full detail. Complete pseudocode including

Xavier initialisation, dropout ($p = 0.1$), gradient flow notes, and memory analysis ($\approx 160$ MB per batch item). Extended version of Algorithm 2.
(PDF)

## Acknowledgments

We thank the volunteers and patients who participated in this study, the nursing staff for clinical data collection support, and the anonymous reviewers for their constructive feedback.

## Author contributions

**Conceptualization:** Oussama El Othmani, Sami Naouali.

**Data curation:** Oussama El Othmani.

**Formal analysis:** Oussama El Othmani.

**Funding acquisition:** Sami Naouali.

**Investigation:** Oussama El Othmani.

**Methodology:** Oussama El Othmani, Sami Naouali.

**Project administration:** Oussama El Othmani.

**Resources:** Sami Naouali.

**Software:** Oussama El Othmani.

**Supervision:** Sami Naouali.

**Validation:** Oussama El Othmani, Sami Naouali.

**Writing – original draft:** Oussama El Othmani, Sami Naouali.

**Writing – review & editing:** Oussama El Othmani, Sami Naouali.

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
