## [Decision Letter · Decision Letter 0]

11 Mar 2026

PDIG-D-26-00203Cross-Spectral Fusion of Thermal and RGB Imaging for Objective Pain EstimationPLOS Digital Health Dear Dr. Naouali, Thank you for submitting your manuscript to PLOS Digital Health. After careful consideration, we feel that it has merit but does not fully meet PLOS Digital Health's publication criteria as it currently stands. Therefore, we invite you to submit a revised version of the manuscript that addresses the points raised during the review process. Please submit your revised manuscript by May 10 2026 11:59PM. If you will need more time than this to complete your revisions, please reply to this message or contact the journal office at digitalhealth@plos.org.  Please include the following items when submitting your revised manuscript:* A letter that responds to each point raised by the editor and reviewer(s). You should upload this letter as a separate file labeled 'Response to Reviewers'. This file does not need to include responses to any formatting updates and technical items listed in the 'Journal Requirements' section below.* A marked-up copy of your manuscript that highlights changes made to the original version. You should upload this as a separate file labeled 'Revised Manuscript with Track Changes'.* An unmarked version of your revised paper without tracked changes. You should upload this as a separate file labeled 'Manuscript'. If you would like to make changes to your financial disclosure, competing interests statement, or data availability statement, please make these updates within the submission form at the time of resubmission. Guidelines for resubmitting your figure files are available below the reviewer comments at the end of this letter. We look forward to receiving your revised manuscript. Kind regards, Jia-Lang Xu

Academic EditorPLOS Digital Health Henry Horng-Shing LuSection EditorPLOS Digital Health Leo Anthony CeliEditor-in-ChiefPLOS Digital Healthorcid.org/0000-0001-6712-6626 **Journal Requirements:**

1. Please ensure that your Ethics Statement is available in its entirety at the beginning of your Methods section, under a subheading 'Ethics Statement'.

2. We ask that a manuscript source file is provided at Revision. Please upload your manuscript file as a .doc, .docx, .rtf or .tex.

3. Please provide an Author Summary. This should appear in your manuscript between the Abstract (if applicable) and the Introduction, and should be 150–200 words long. The aim should be to make your findings accessible to a wide audience that includes both scientists and non-scientists. Sample summaries can be found on our website under Submission Guidelines:

https://journals.plos.org/digitalhealth/s/submission-guidelines#loc-parts-of-a-submission

4. Please ensure that all Figure files have corresponding citations and legends within the manuscript. Currently, Figures 12, 13, and 14 in your submission file inventory does not have an in-text citation. If the figure is no longer to be included as part of the submission, please remove it from the file inventory.

5. We have noticed that you have uploaded Supporting Information files, but you have not included a list of legends. Please add a full list of legends for your Supporting Information files after the references list.

6. In the online submission form, you indicated that “Code: The source code for our CSAF+Transformer model, preprocessing pipelines,

and evaluation scripts are available from the corresponding author upon reasonable

request: https://github.com/oussama123-ai/Cross-Spectral-Fusion-of-Thermal.

Trained Models: Pre-trained model weights will be released via the same

repository, enabling reproduction of our results and facilitating further research.

Data: Due to patient privacy concerns and institutional review board restrictions,

we cannot publicly release raw video recordings. However, we will provide:

• Processed Features: Anonymized regional features extracted from thermal and

RGB modalities (no identifiable facial images) for research use

• Metadata: Dataset statistics, pain intensity distributions, demographic informa tion (aggregated to prevent identification)

• Evaluation Protocol: Exact train/validation/test splits, cross-validation fold

definitions, and evaluation scripts for reproducibility

Researchers wishing to access processed features must submit a data use agree ment to the corresponding author detailing intended use, privacy protection measures,

and commitment to not attempt re-identification. Requests will be reviewed by our

institutional data governance committee and approved within 30 days if appropriate

protections are in place.”

3. Uploaded as supplementary information.

7. Some material included in your submission may be copyrighted. According to PLOS’s copyright policy, authors who use figures or other material (e.g., graphics, clipart, maps) from another author or copyright holder must demonstrate or obtain permission to publish this material under the Creative Commons Attribution 4.0 International (CC BY 4.0) License used by PLOS journals. Please closely review the details of PLOS’s copyright requirements here: PLOS Licenses and Copyright. If you need to request permissions from a copyright holder, you may use PLOS's Copyright Content Permission form.

Potential Copyright Issues:

a. Figure 1: Please confirm whether you drew the images / clip-art within the figure panels by hand. If you did not draw the images, please provide (a) a link to the source of the images or icons and their license / terms of use; or (b) written permission from the copyright holder to publish the images or icons under our CC-BY 4.0 license. Alternatively, you may replace the images with open source alternatives. See these open source resources you may use to replace images / clip-art:

- https://openclipart.org/

 If the reviewer comments include a recommendation to cite specific previously published works, please review and evaluate these publications to determine whether they are relevant and should be cited. There is no requirement to cite these works unless the editor has indicated otherwise.  **Additional Editor Comments (if provided):** This manuscript has been under review for a relatively long time, and we hope this delay has not affected your willingness to submit to this journal. Two reviewers have already provided their comments. Please respond to the issues raised by the reviewers.**Reviewers' Comments:** Reviewer's Responses to Questions

**Comments to the Author**

1. Does this manuscript meet PLOS Digital Health’s publication criteria? Is the manuscript technically sound, and do the data support the conclusions? The manuscript must describe methodologically and ethically rigorous research with conclusions that are appropriately drawn based on the data presented.

Reviewer #1: Yes

Reviewer #2: Partly

2. Has the statistical analysis been performed appropriately and rigorously?

Reviewer #1: Yes

Reviewer #2: N/A

3. Have the authors made all data underlying the findings in their manuscript fully available (please refer to the Data Availability Statement at the start of the manuscript PDF file)?

Reviewer #1: Yes

Reviewer #2: Yes

4. Is the manuscript presented in an intelligible fashion and written in standard English?

Reviewer #1: Yes

Reviewer #2: Yes

5. Review Comments to the Author

**Reviewer #1:** Overview:

Authors have developed a new cross-attention fusion model using thermal imaging and RBG facial analysis and have conducted convincing experiments on the approach with two in-human datasets (one with experimentally induced pain, and one with real clinical post-operative surgical pain). This work is a good initial basic science/feasibility work. As authors mention, further validation is needed for target applications in neonates, cognitively impaired individuals, and sedated patients. Notably, for many of these patient populations, facial expressions may be atypical (e.g. cognitively impaired individuals) or absent (e.g. sedated patients or patients under anesthesia). Therefore, validation studies in these target use cases require evaluation of the model either in the absence of RGB video or with normal facial expressions in RGB video. My main concern is that the manuscript is too verbose and should be written in a more focused manner that more clearly conveys the authors’ modeling and experimental contributions.

Strengths:

1. Deep learning architecture with cross-attension fusion of thermal imaging and RBG imaging is a good idea that is physiologically grounded and worth exploring

2. Validation on 2 types of datasets (induced pain vs. post-operative surgical pain) which were collected by research team to form novel thermal + RBG video datasets in the pain literature.

3. Comparison against many baseline models which represent diverse modeling approaches.

4. Comparison against human experts (3 experienced nurses on subset of data)

5. Results are reported in great detail with appropriate statistical testing, ablation studies, and clear linkage to hypotheses posed by authors.

6. Interesting finding that thermal imaging has better resolution/accuracy at resolving pain scores for high pain intensity.

7. Great analysis on how thermal & RBG modalities have distinct contributions using learned adaptive gates and localized by pain intensity, expression, and facial region (Table 6).

Major Concerns/Comments:

1. General: This manuscript is too long and reads more like a textbook chapter or review article rather than an original investigation research manuscript. It is roughly 13,000 words whereas most research manuscripts are typically in the range of 3,000-6,000 words. Almost every section contains too much detail. Same/similar information is restated across different sections. Excessive background detail buries the research team’s main experimental contributions. Consider splitting some of the manuscript’s elaboration on historical background, implementation details such as technical constraints, deployment barriers, discussion, etc. into a supplementary appendix.

2. Introduction: The introduction/background/research vision/contributions is also too long spanning 4 pages (many research articles are only 4-6 pages in length). Please significantly shorten this to less to one page, focused around the research vision and hypothesis. The contributions/significance subsection ends up repeating the results section. Only briefly mention contributions at the end of the introduction section or move to the results section entirely. Elaboration of contributions at the end of introduction is more common in CS conference manuscripts rather than health/clinical journal manuscripts.

3. Introduction: The related work section needs to either be incorporated into introduction or discussion as it is not one of the sections of manuscript organization. This section makes the manuscript read more like a review article rather than a manuscript focused on new research methodology. If the goal is to review the state-of-the-art of the field, please write a separate review article with the content. For this article, significantly trim and focus the article on your novel contributions & experiments.

4. Methods: Your Table 1 is a comparison against prior research. This is appropriate for a review article but not for a manuscript focusing on original research investigation. If you want to include this table, make it a supplemental table that summarizes existing literature. Commonly, Table 1 for original research articles in clinical medicine is often part of the results and includes a description of dataset/research cohort feature & outcome variables as well as patient population factors (e.g. age, sex, relevant comorbidities, etc.).

5. Results: Authors summarize additional analyses on cross-dataset generalization and human expert comparison, but do not report details. These are two very important areas of results that affect the likelihood of actual clinical use & clinical adoption and should be included/emphasized in greater detail rather than an afterthought.

Minor Concerns/Comments:

1. Abstract: Wording combines the 2 separate datasets for controlled pain induction vs. clinical postoperative monitoring, which are different. I would not combine these as they test different mechanisms/categories in pain research (induced discomfort vs. surgical pain). Furthermore, induced pain is pre-clinical testing whereas clinical post-operative monitoring of surgical pain is real-world validation in clinical use and the dataset targets and use cases are different. By combining, the authors assume that all types of pain are similar/same and have similar patterns of pain response which is not true.

2. Methods: Your Table 1 is a comparison against prior research. This is appropriate for a review article but not for a manuscript focusing on original research methodology. If you want to include this table, make it a supplemental table that summarizes existing literature.

3. Table 2: Explain abbreviation such as “NRS” in captions.

4. Table 4 and Figure 3 mostly show the same information–please consolidate. If you choose to keep the plots in Figure 3, please add error bars/confidence intervals.

5. Methods/Results: consider selecting one of the metrics as your “primary” metric and the others as “secondary” to help readers focus on a main metric for comparison.

6. The focus of this paper is to analyze and validate cross-modal fusion thermal + RBG deep learning models. Authors then suggest that the technology could be useful for low expressors or patients who are sedated where patients have limited facial change and presumably may have thermal changes, but this data distribution is out-of-domain compared to the examined datasets the authors study (patients can have dynamic facial expressions and thermal changes). The thermal + RBG models and approach would have to be validated specifically against this patient population as future work as the current proof-of-concept work can suggest utility, but is not directly translatable to the low expressor & sedated patient population. It is more likely the current work has greater applicability to pediatric and neonatal populations who cannot verbally express pain, but can have facial expressions and thermal changes. However, pediatric and neonatal facial expressions and physiology is different from adults, so separate validation work needs to be performed for this use case as well–the authors appropriately state that additional validation is needed for this use case.

7. PLOS Data policy requires all data be made available with rare exceptions. In the data availability statement, authors make available code and processed features but cannot make available facial images/video due to protected health information (PHI) and IRB concerns, which I believe to be reasonable and qualifies for the rare exception.

**Reviewer #2:** The Introduction section contains overly lengthy paragraphs; it would be beneficial to make this section more concise. In addition, the research objectives should be stated more clearly to better highlight the purpose and contribution of the study. A similar issue is observed in the Related Work section, where the descriptions are also excessively long and could be streamlined.

In Table 1, it appears that previous studies have primarily reported performance using accuracy as the evaluation metric. The authors should clarify why MAE is used as the primary metric in this manuscript. The description of the Datasets should also be presented more clearly, particularly regarding the final dataset size used in the study.

In Table 3, since the study employs 5-fold cross-validation, it is recommended to report each evaluation metric together with its error range (e.g., standard deviation or confidence interval) to provide a more comprehensive assessment of model performance. The explanations for all figures and tables should also be clarified to improve readability and interpretation.

6. PLOS authors have the option to publish the peer review history of their article (what does this mean?). If published, this will include your full peer review and any attached files.

**Do you want your identity to be public for this peer review?** For information about this choice, including consent withdrawal, please see our Privacy Policy.

Reviewer #1: **Yes:** Philip Chung

Reviewer #2: **Yes:** Pei-Chun Lin

  **Figure resubmission:** While revising your submission, we strongly recommend that you use PLOS’s NAAS tool (https://ngplosjournals.pagemajik.ai/artanalysis) to test your figure files. NAAS can convert your figure files to the TIFF file type and meet basic requirements (such as print size, resolution), or provide you with a report on issues that do not meet our requirements and that NAAS cannot fix.

After uploading your figures to PLOS’s NAAS tool - https://ngplosjournals.pagemajik.ai/artanalysis, NAAS will process the files provided and display the results in the "Uploaded Files" section of the page as the processing is complete. If the uploaded figures meet our requirements (or NAAS is able to fix the files to meet our requirements), the figure will be marked as "fixed" above. If NAAS is unable to fix the files, a red "failed" label will appear above. When NAAS has confirmed that the figure files meet our requirements, please download the file via the download option, and include these NAAS processed figure files when submitting your revised manuscript. **Reproducibility:** To enhance the reproducibility of your results, we recommend that authors of applicable studies deposit laboratory protocols in protocols.io, where a protocol can be assigned its own identifier (DOI) such that it can be cited independently in the future. Additionally, PLOS ONE offers an option to publish peer-reviewed clinical study protocols. Read more information on sharing protocols at https://plos.org/protocols?utm_medium=editorial-email&utm_source=authorletters&utm_campaign=protocols

---

## [Decision Letter · Decision Letter 1]

6 Apr 2026

PDIG-D-26-00203R1Cross-Spectral Fusion of Thermal and RGB Imaging for Objective Pain EstimationPLOS Digital Health Dear Dr. Naouali, Thank you for submitting your manuscript to PLOS Digital Health. After careful consideration, we feel that it has merit but does not fully meet PLOS Digital Health's publication criteria as it currently stands. Therefore, we invite you to submit a revised version of the manuscript that addresses the points raised during the review process. Please submit your revised manuscript by Jun 05 2026 11:59PM. If you will need more time than this to complete your revisions, please reply to this message or contact the journal office at digitalhealth@plos.org.  Please include the following items when submitting your revised manuscript:* A letter that responds to each point raised by the editor and reviewer(s). You should upload this letter as a separate file labeled 'Response to Reviewers'. This file does not need to include responses to any formatting updates and technical items listed in the 'Journal Requirements' section below.* A marked-up copy of your manuscript that highlights changes made to the original version. You should upload this as a separate file labeled 'Revised Manuscript with Track Changes'.* An unmarked version of your revised paper without tracked changes. You should upload this as a separate file labeled 'Manuscript'. If you would like to make changes to your financial disclosure, competing interests statement, or data availability statement, please make these updates within the submission form at the time of resubmission. Guidelines for resubmitting your figure files are available below the reviewer comments at the end of this letter. We look forward to receiving your revised manuscript. Kind regards, Jia-Lang XuAcademic EditorPLOS Digital Health Jia-Lang XuAcademic EditorPLOS Digital Health Leo Anthony CeliEditor-in-ChiefPLOS Digital Healthorcid.org/0000-0001-6712-6626 **Journal Requirements:** If the reviewer comments include a recommendation to cite specific previously published works, please review and evaluate these publications to determine whether they are relevant and should be cited. There is no requirement to cite these works unless the editor has indicated otherwise.  **Additional Editor Comments (if provided):** This manuscript has received additional comments from reviewers during this revision process; please address these comments accordingly.**Reviewers' Comments:** Reviewer's Responses to Questions

**Comments to the Author**

1. If the authors have adequately addressed your comments raised in a previous round of review and you feel that this manuscript is now acceptable for publication, you may indicate that here to bypass the “Comments to the Author” section, enter your conflict of interest statement in the “Confidential to Editor” section, and submit your "Accept" recommendation.

Reviewer #1: All comments have been addressed

Reviewer #2: (No Response)

2. Does this manuscript meet PLOS Digital Health’s publication criteria? Is the manuscript technically sound, and do the data support the conclusions? The manuscript must describe methodologically and ethically rigorous research with conclusions that are appropriately drawn based on the data presented.

Reviewer #1: Yes

Reviewer #2: (No Response)

3. Has the statistical analysis been performed appropriately and rigorously?

Reviewer #1: Yes

Reviewer #2: (No Response)

4. Have the authors made all data underlying the findings in their manuscript fully available (please refer to the Data Availability Statement at the start of the manuscript PDF file)?

Reviewer #1: Yes

Reviewer #2: (No Response)

5. Is the manuscript presented in an intelligible fashion and written in standard English?

Reviewer #1: Yes

Reviewer #2: (No Response)

6. Review Comments to the Author

Reviewer #1: The manuscript reads much better and is substantially improved. In particular, I appreciate the addition of [Sec sec026] (Cross Dataset Generalization) as well as 4.11 (Human Expert Comparison). The Human Expert Comparison data is convincing and shows real clinical utility of the RBG+Thermal imaging system given the high ICC achieved when compared to current standard of care with nurse assessment while achieving tighter confidence interval bounds for the primary MAE metric.

My remaining concern pertains to interpretation of Fig 7 & Fig 8. It is unclear how to interpret spatial attention maps without facial image context. The captions claim attention is focused on specific parts of the face, but looking at the heatmaps alone, I cannot tell if there is a face in the heat map and where the facial features would be located. To assist the reader in interpreting these figures while protecting identities of original participants, I recommend superimposing a representative “average” sketched face diagram on the heatmap. This would allow the reader to link the heatmaps to specific facial regions.

Reviewer #2: Although this manuscript has undergone substantial revisions, several issues still require further improvement:

The Introduction section remains too brief. Please expand it to better highlight the novelty and significance of this study.

The descriptions of the tables and figures are overly concise and should be elaborated to improve clarity and interpretability.

Figure 7 is a critical component of this manuscript. Please strengthen the analysis in this section and identify the common patterns or key insights derived from these visualizations.

The Discussion section is still insufficient. Please expand it by elaborating on the study findings and providing a more thorough comparison with existing literature.

7. PLOS authors have the option to publish the peer review history of their article (what does this mean?). If published, this will include your full peer review and any attached files.

**Do you want your identity to be public for this peer review?** For information about this choice, including consent withdrawal, please see our Privacy Policy.

Reviewer #1: **Yes:** Philip Chung

Reviewer #2: **Yes:** Pei-Chun Lin

  **Figure resubmission:**  While revising your submission, we strongly recommend that you use PLOS’s NAAS tool (https://ngplosjournals.pagemajik.ai/artanalysis) to test your figure files. NAAS can convert your figure files to the TIFF file type and meet basic requirements (such as print size, resolution), or provide you with a report on issues that do not meet our requirements and that NAAS cannot fix. 

After uploading your figures to PLOS’s NAAS tool - https://ngplosjournals.pagemajik.ai/artanalysis, NAAS will process the files provided and display the results in the "Uploaded Files" section of the page as the processing is complete. If the uploaded figures meet our requirements (or NAAS is able to fix the files to meet our requirements), the figure will be marked as "fixed" above. If NAAS is unable to fix the files, a red "failed" label will appear above. When NAAS has confirmed that the figure files meet our requirements, please download the file via the download option, and include these NAAS processed figure files when submitting your revised manuscript. **Reproducibility:** To enhance the reproducibility of your results, we recommend that authors of applicable studies deposit laboratory protocols in protocols.io, where a protocol can be assigned its own identifier (DOI) such that it can be cited independently in the future. Additionally, PLOS ONE offers an option to publish peer-reviewed clinical study protocols. Read more information on sharing protocols at https://plos.org/protocols?utm_medium=editorial-email&utm_source=authorletters&utm_campaign=protocols

---

## [Decision Letter · Decision Letter 2]

23 Apr 2026

Cross-Spectral Fusion of Thermal and RGB Imaging for Objective Pain Estimation

PDIG-D-26-00203R2

Dear Prof Naouali,

We are pleased to inform you that your manuscript 'Cross-Spectral Fusion of Thermal and RGB Imaging for Objective Pain Estimation' has been provisionally accepted for publication in PLOS Digital Health.

Best regards,

Jia-Lang Xu

Academic Editor

PLOS Digital Health

**Additional Editor Comments (if provided):**

This manuscript has adequately addressed the reviewers’ comments during the review process and is suitable for publication.

**Reviewer Comments (if any, and for reference):**

Reviewer's Responses to Questions

**Comments to the Author**

1. If the authors have adequately addressed your comments raised in a previous round of review and you feel that this manuscript is now acceptable for publication, you may indicate that here to bypass the “Comments to the Author” section, enter your conflict of interest statement in the “Confidential to Editor” section, and submit your "Accept" recommendation.

Reviewer #1: All comments have been addressed

Reviewer #2: All comments have been addressed

2. Does this manuscript meet PLOS Digital Health’s publication criteria? Is the manuscript technically sound, and do the data support the conclusions? The manuscript must describe methodologically and ethically rigorous research with conclusions that are appropriately drawn based on the data presented.

Reviewer #1: Yes

Reviewer #2: Yes

3. Has the statistical analysis been performed appropriately and rigorously?

Reviewer #1: Yes

Reviewer #2: Yes

4. Have the authors made all data underlying the findings in their manuscript fully available (please refer to the Data Availability Statement at the start of the manuscript PDF file)?

Reviewer #1: Yes

Reviewer #2: Yes

5. Is the manuscript presented in an intelligible fashion and written in standard English?

Reviewer #1: Yes

Reviewer #2: Yes

6. Review Comments to the Author

Reviewer #1: Thank you for addressing my concerns.

Reviewer #2: None

7. PLOS authors have the option to publish the peer review history of their article (what does this mean?). If published, this will include your full peer review and any attached files.

**Do you want your identity to be public for this peer review?** For information about this choice, including consent withdrawal, please see our Privacy Policy.

Reviewer #1: **Yes:** Philip Chung

Reviewer #2: **Yes:** Pei-Chun Lin
